# Evaluating fomite risk of brown paper bags storing personal protective equipment exposed to SARS-CoV-2: A quasi-experimental study

Kyirsty Unger[1]*, Leslie Dietz[2], Patrick Horve[2], Kevin Van Den Wymelenberg[2], Amber Lin[1], Erin Kinney[3], Bory Kea[4]

1 Oregon Health & Science University, Portland, Oregon, United States of America, 2 University of Oregon, Eugene, Oregon, United States of America, 3 School of Medicine, Oregon Health & Science University, Portland, Oregon, United States of America, 4 Department of Emergency Medicine, Center for Policy and Emergency Medicine, Oregon Health & Science University, Portland, Oregon, United States of America

* kyirsty@gmail.com

## Abstract

### Introduction

Literature is lacking on the safety of storing contaminated PPE in paper bags for reuse, potentially increasing exposure to frontline healthcare workers (HCW) and patients. The aim of this study is to evaluate the effectiveness of paper bags as a barrier for fomite transmission of SARS-CoV-2 by storing face masks, respirators, and face shields.

### Methods

This quasi-experimental study evaluated the presence of SARS-CoV-2 on the interior and exterior surfaces of paper bags containing PPE that had aerosolized exposures in clinical and simulated settings. Between May and October 2020, 30 unique PPE items were collected from COVID-19 units at two urban hospitals. Exposed PPE, worn by either an infected patient or HCW during a SARS-CoV-2 aerosolizing event, were placed into an unused paper bag. Samples were tested at 30-minute and 12-hour intervals.

### Results

A total of 177 swabs were processed from 30 PPE samples. We found a 6.8% positivity rate among all samples across both collection sites. Highest positivity rates were associated with ventilator disconnection and exposure to respiratory droplets from coughing. Positivity rates differed between hospital units. Total positivity rates were similar between 30-minute (6.7%) and 12-hour (6.9%) sample testing time intervals. Control samples exposed to inactivated SARS-CoV-2 droplets had higher total viral counts than samples exposed to nebulized aerosols.

**Data Availability Statement:** All relevant data are within the manuscript and its Supporting Information files.

**Funding:** This study was funded by Oregon Clinical and Translational Research Institute. The funders had no role in the study design, data collection, and analysis, decision to publish, or preparation of the manuscript.

**Competing interests:** Van Den Wymelenberg has a company called Duktile through which he provides healthy building consulting, including consulting related to viral pathogens, and he serves as a scientific advisor to EnviralTech, a company that conducts viral environmental surveillance, including in senior care facilities. This does not alter our adherence to PLOS ONE policies on sharing data and materials.

## Conclusions

Data suggests paper bags are not a significant fomite risk for SARS-CoV-2 transmission. However, controls demonstrated a risk with droplet exposure. Data can inform guidelines for storing and re-using PPE in situations of limited supplies during future pandemics.

## Introduction

Severe Acute Respiratory Syndrome Coronavirus 2 (SARS-CoV-2), the causative agent of coronavirus disease 2019 (COVID-19), presents a significant exposure risk to health care workers (HCWs) globally [1,2]. Limited global distribution and manufacturing capacity of personal protective equipment (PPE) prompts the recommendation of circumstantial guidelines to address limited supply [3]. One strategy implemented permits the reuse of disposable PPE items. The Centers for Disease Control and Prevention (CDC) guidelines for procedures, limitations, and storage of reused disposable PPE recommended storing reused PPE in paper bags [3]. These guidelines do not address the decontamination of bags or specify safe handling and storage considerations.

PPE poses fomite transmission risk for HCWs with both prolonged use and after donning and doffing off reused items [4–6]. Prolonged use is defined as any application of PPE worn continuously beyond standard patient care or greater than 1 hour of continuous use [7]. Disposable PPE, such as surgical masks, have been found to have a significant viral and bacterial burden with extended use (over 4–6 hours of continuous use) [8] on interior and exterior surfaces of the mask. Prolonged use also demonstrated increases of transmission via surgical site infections (SSI) [4,8]. COVID-19 Transmission is primary caused by droplet-borne and short-range airborne exposures to exhaled particles [9]. Initial studies found that SARS-CoV-2 was detectable on fomites for up to seventeen days [10,11] and found to be more stable on plastic than on cardboard, with no viable virus detected on cardboard after 24 hours, while plastic retained detectable virus up to 72 hours [8].

While the use of paper bags as a PPE containment strategy had not been previously documented, the CDC was guided by previous studies showing that porous surfaces demonstrated a lower transfer efficiency than most non-porous surfaces for bacterial contaminants and most viral transmissions [10,12]. Paper bags are porous, inexpensive, and often readily accessible. As a result, they can easily be implemented in healthcare settings as storage for PPE during an epidemic or pandemic.

Although paper bags are easily accessible, the vulnerability for fomite transmission from reused PPE items to other surfaces such as paper bags, is unknown. Given that HCWs access these materials multiple times within a single workday shift, there is a potential transmission risk from fomite exposure [13] with the handling of paper bags without proper protection or evidence-based guidelines [3]. Our study addresses this concern by evaluating for the presence of SARS-CoV-2 on interior and exterior surfaces of paper bags containing PPE exposed in both clinical and laboratory settings. These findings can better inform operational processes of resource limited settings.

## Methods

### Study design

This is a quasi-experimental study. Samples were collected from two regional urban hospitals in the Portland metropolitan area. Patients were identified by electronic health record and

inclusion criteria were verified by staff caring for the identified patients (see Sample Collection). Inclusion criteria included PPE from HCWs exiting a room of a patient who had a positive COVID-19 polymerase chain reaction (PCR) test *and* was currently undergoing an aerosolizing event [13]. Aerosolizing events, also known as aerosol-generating procedures (AGPs) [13], could include open suctioning of airways, sputum induction, endotracheal intubation and extubation, or the use of a BiPAP or CPAP [3,13,14]. Use of a high-flow nasal cannula was also included as an AGP for this study. Surgical masks (ATSM level 3 mask) [15] worn for over 20 minutes by a symptomatic hospitalized patient with a positive COVID-19 PCR test were included. A COVID-symptomatic patient was defined as exhibiting one or more of the following: shortness of breath, fever or frequent coughing [16].

At the time of this study, PPE was limited to continuous use at each of these healthcare institutions. The policy for continuous PPE use at both institutions involved a healthcare worker receiving a new N95 or KN95 [15] mask at the beginning of their shift and wearing it continuously throughout their 12-hour shift. Plastic face shields were worn inside rooms containing a COVID-19 patient. All PPE was doffed and cleaned according to safety procedures, and then stored and reused by the same user if needed. Gowns were doffed and discarded. Face shields and eye protection were cleaned with Oxivir. The PPE item to be sampled was placed in a clean brown paper bag and placed in a designated sampling area. For HCWs workflow convenience, samples were collected by screening all staff in the designated department at shift change times (0700 and 1900) by four trained researchers and followed sampling methods and criteria described. After obtaining a PPE sample, researchers asked the primary nurse to provide the date of the patient's last positive COVID-19 PCR test and compared it to the sampling date.

The Institutional Review Board (IRB) determined this study to be exempt for both hospital locations. Protected patient data was not recorded, and a waiver of consent was obtained.

## Sample collection

Between May and October 2020, 30 unique PPE items were collected from critical and intermediate care COVID-19 units at two urban hospitals—an academic trauma center and a community hospital. All PPE tested was exposed to SARS-CoV-2 worn by either an infected patient (ASTM level 3 mask) [15] or healthcare provider (N95 or face shield) [15] during an aerosolizing event were collected from outside the patient room and placed into an unused 16-pound paper bag (Fig 1). The HCW was observed doffing disposable gowns and gloves in the patient room and then performing hand hygiene before placing the exposed PPE in an unused paper bag. The paper bag was then labeled inconspicuously so as not to interfere with the ongoing experiment. Researchers also wore all appropriate PPE per hospital safety guidelines before handling the paper bags containing exposed PPE. Each exposed PPE item was given a unique identification number and a recorded collection time.

The paper bags were then taken from the hospital unit for storage and swab collection. A designated, secure, non-patient area was disinfected with Oxivir (Diversey, Catalog #1008050923). This sampling area was swabbed for environmental contamination. Sampling occurred by moving a pre-moistened flocked swab (Copan Diagnostics, Catalog # COP-520CS01) in a back-and-forth pattern across the 25cm sq. designated sampling area for approximately 20 seconds. All bags and PPE were sampled 30 minutes after the exposed PPE item was placed in the bag. Next, the bag's interior and corresponding exterior surface, in contact with the exposed PPE item, were sampled. Sampling was repeated 12-hours after the initial placement of the exposed PPE in the bag. At 12-hours, the unexposed interior surface of the exposed PPE item was sampled, and then the interior and exterior surfaces of the paper bags in contact with the exposed side of the exposed PPE were sampled. Researchers swabbed a different,

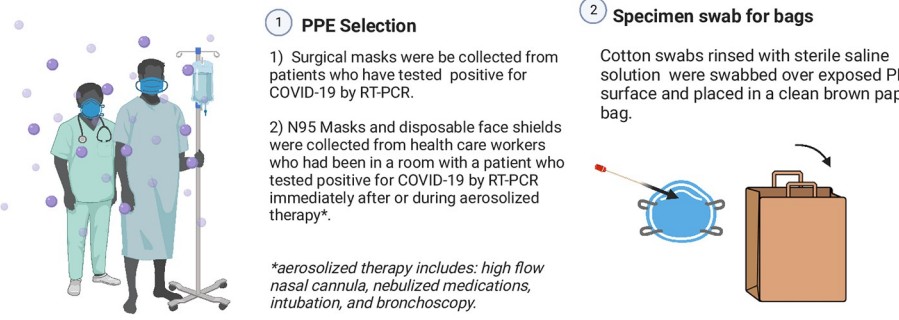

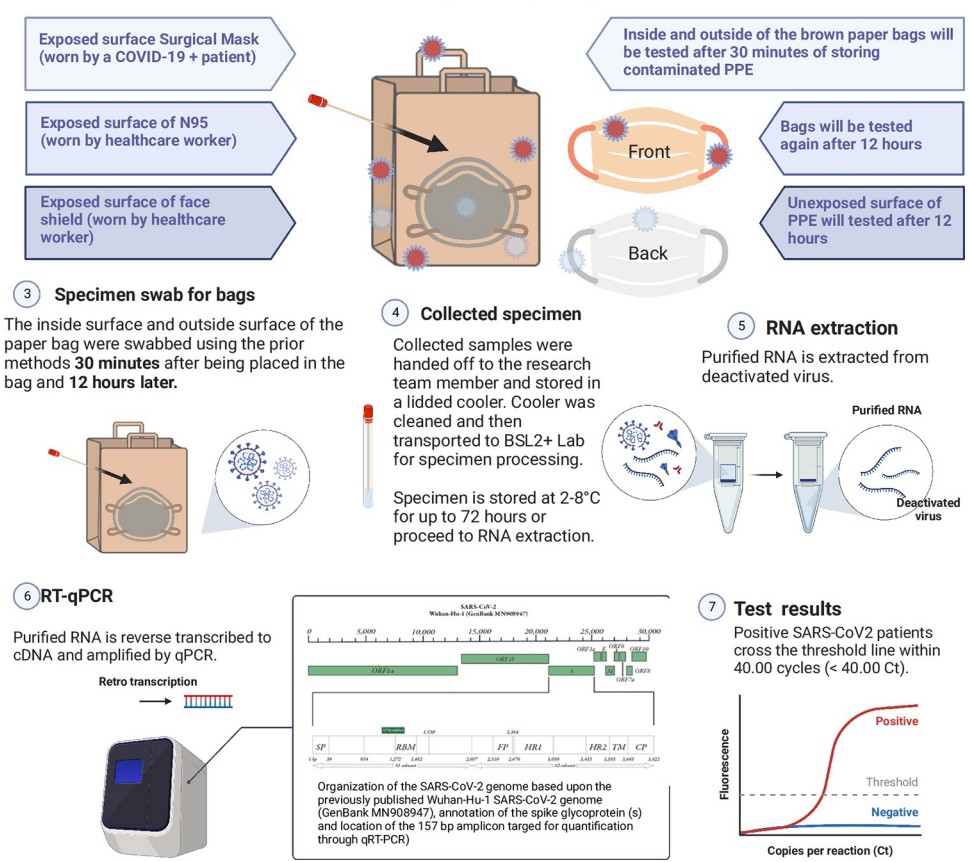

**Fig 1. Protocol Infographic "Created with BioRender.com".**

though similarly representative, area of the bag that had been sampled at the 30-minute time point sampling. After the sampling period was concluded, samples were transported on ice to a BSL-2+ (enhanced safety precautions) laboratory for initial processing. Samples were briefly vortexed, allowed to settle for a 5-minute period, and 200 μl of the supernatant was removed, and combined with 600 μl of a lysis/preservative buffer (DNA/RNA Shield, Zymo Research #R1100). This work took place in a class 2 biosafety cabinet (BSC). Samples were then transported by motor vehicle to a BSL-2 laboratory for further processing and analysis.

## Sample processing and molecular analysis

Total RNA was extracted using the Quick RNA viral kit (Zymo Research #R1035) and stored at -80˚C until quantitative polymerase chain reaction (qPCR) could be conducted. As described by Chan et al. 2020, the quantity of SARS-CoV-2 of each sample was assessed by qPCR using specific real-time qPCR methods targeting a 157 bp segment of the S1 subunit of the spike glycoprotein (S) gene of the SARS-CoV-2 [17]. All samples were run in triplicate to validate the empirical data.

## Controls

Clean paper bags were obtained from the hospital supply. Uniform 2-inch square pieces of clean paper bag were placed in room-scale controlled environment with an internal volume of 28,040 L of air. A known concentration of inactivated SARS-CoV-2 was nebulized such that approximately 3,180 gene copies per L of room air were achieved at steady state aerosol concentration within the room to simulate a high load SARS-CoV-2 aerosolization event. Virus was aerosolized using three 4-jet Blaustein Atomizing Modules (CH Technologies) each with a flow rate of 16 L/min at 50 psi/. Droplet events were simulated by placing 2 μl (approximately 150,000 gene copies) of inactivated SARS-CoV-2 by pipette on pieces of clean paper bag in a biosafety cabinet. Negative control trials were always conducted prior to experimental trials to ensure there is no carry-over contamination. Paper bag pieces were swabbed at time intervals of 30 minutes and 12 hours to mimic the healthcare clinical setting experiment. These control samples were performed in triplicate to validate the empirical data. Additionally, one set of paper bag pieces were not exposed to any virus and swabbed at the same time intervals to serve as a negative control. The PCR reaction contained control samples from each RNA extraction batch as well as a no-template control (NTC) to rule out possible laboratory contamination.

In the control experiment, a known concentration of heat-inactivated SARS-CoV-2 was nebulized. This was prepared from a concentrated stock that had been quantified by the BSL3 lab at Montana State University where it was cultivated and attenuated and then verified by qPCR by our lab upon arrival.

The control experiment, nebulization were conducted in a sealed room-scale controlled environment with an internal volume of 28,040 L with an internal temperature maintained at 22˚C +/- 4˚C and a relative humidity maintained at 50% +/- 10% with the use of a single portable humidifier. Air was circulated in the room using two oscillating fans, which moved 24,975 L of air per minute [18]. During this experiment, the air exchange rate was 1 ACH and was regulated and sustained using a timed operation HEPA exhaust, with make up air via infiltration [18].

## Statistical analysis

We calculated the proportion of positive tests overall and by PPE type, exposure type, and location. We then calculated 95% confidence interval (CI) for the proportion of positive tests using the Agresti-Coull method [19]. Analysis was performed in Stata 16 (College Station, TX; StataCorp LLC).

## Results

### Clinical setting

In total, 30 exposed PPE samples were collected and tested at 30-minute and 12-hour intervals resulting in 177 qPCR tests (Table 1). The distribution of exposed PPE items sampled is included in Table 1. Of the 30 unique exposed PPE items, nine were found to have positive qPCR tests (30%) at one point during the sampling process (Table 2). We found a 6.8%

**Table 1. Samples by area swabbed and personal protective equipment (PPE) type.**

| *Distribution of samples by area and PPE type* | | | | |
| --- | --- | --- | --- | --- |
| *Sample area* | PPE Type | | | |
| | **Face Shield** | **N95** | **Surgical Mask** | **Total** |
| *Inside of Bag* | 6 | 40 | 13 | 59 |
| *Inside of Mask/Shield* | 6 | 28 | 13 | 47 |
| *Outside of Bag* | 6 | 40 | 13 | 59 |
| *Outside of Mask/Shield* | 0 | 12 | 0 | 12 |
| *Total Samples* | 18 | 120 | 39 | 177 |

**Table 2. Proportion of positive tests.**

| *Exposure Type* | *Number of samples* | *Percent positive tests (95% CI[*])* |
| --- | --- | --- |
| *Total* | 177 | 6.8%(3.8% - 11.6%) |
| *Sample area* | | |
| *Inside of bag* | 59 | 6.8%(2.2% - 16.6%) |
| *Inside of mask/shield* | 47 | 6.4%(1.6% - 17.8%) |
| *Outside of bag* | 59 | 8.5%(3.3% - 18.8%) |
| *Outside of mask* | 12 | 0.0%(0.0% - 28.2%) |
| *PPE type* | | |
| *Face shield* | 18 | 0.0%(0.0% - 20.7%) |
| *N95* | 120 | 7.5%(3.8% - 13.8%) |
| *Surgical mask* | 39 | 7.7%(1.9% - 21.0%) |
| *Time period* | | |
| *30 minutes* | 90 | 6.7%(2.8% - 14.1%) |
| *12 hours* | 87 | 6.9%(2.9% - 14.5%) |
| *Exposure type* | | |
| *Direct cough* | 24 | 8.3%(1.2% - 27.0%) |
| *High flow nasal cannula* | 129 | 6.2%(3.0% - 11.9%) |
| *Tracheostomy surgery* | 18 | 5.6%(0.0% - 27.6%) |
| *Ventilator disconnect* | 6 | 16.7%(1.1% - 58.2%) |
| *Location* | | |
| *Trauma Center: Medical step-down COVID-19 unit* | 6 | 16.7%(1.1% - 58.2%) |
| *Trauma Center: Observation Unit* | 6 | 0.0%(0.0% - 44.3%) |
| *Trauma Center: Emergency Department* | 21 | 4.8%(0.0% - 24.4%) |
| *Trauma Center: COVID-19 ICU* | 114 | 8.8%(4.7% - 15.6%) |
| *Community Hospital: COVID-19 Step down unit* | 30 | 0.0%(0.0% - 13.5%) |
| *Role* | | |
| *Anesthesiology* | 6 | 0.0%(0.0% - 44.3%) |
| *Patient* | 24 | 8.3%(1.2% - 27.0%) |
| *RN* | 111 | 8.1%(4.1% - 14.9%) |
| *RN dialysis* | 6 | 0.0%(0.0% - 44.3%) |
| *RN circulator* | 6 | 0.0%(0.0% - 44.3%) |
| *RT* | 18 | 0.0%(0.0% - 20.7%) |
| *Surgical technician* | 6 | 16.7%(1.1% - 58.2%) |

[*]Agresti-Coull confidence intervals.

positivity rate in the total swabs collected. Exposed PPE were collected primarily from an academic hospital (25 unique exposed PPE samples total) and five unique exposed PPE samples from a community hospital. Most exposed PPE items (n = 111) tested were N95 masks worn by nurses with exposure to aerosolization by a high flow nasal cannula.

We analyzed the percent positive samples for patterns by comparing the sampled area, PPE type, exposure time, and location sampled, and did not find consistencies or discernible patterns within our data (*Table 2*). Within the sampled areas tested, the outside or "exposed side" of the PPE did not yield positive swabs while the inside of the PPE, or the "unexposed" side, and the inside and outside of the paper bag returned positive results. When comparing the types of PPE, the N95/KN95 masks and ASTM type 3 masks had similar rates of qPCR positivity. Of the three unique face shield samples collected, none returned a positive qPCR result. Face shields were the only items collected that had been decontaminated by Oxivir wipe prior to sampling. The time intervals of 30 minutes and 12 hours did not appear to impact the rates of positivity and were 6.7% and 6.9% respectively (*Table 3*).

Slight variation occurred when comparing the type of exposure with the rates of positivity (*Table 2*), with the highest rate being PPE exposed to a disconnected ventilator (16.7%). The interior surfaces of masks worn by coughing COVID-19 patients demonstrated an 8.3% positivity rate. High-flow nasal cannula and bedside tracheostomy demonstrated the lowest rates of positivity, at 6.2% and 5.6%, respectively.

The location of items collected had varying rates of positivity. Most exposed PPE items in this study (64.4%) were collected from an academic trauma center COVID-19 Intensive Care Unit (ICU). The unit with the highest rate of positivity was the academic trauma center, step-down COVID-19 unit with 16.7%. Only ASTM level 3 masks were sampled from this unit due to the hospital policy and workflow (where only N95s or KN95s were worn for AGP events and ASTM level 3 masks were the standard at that time). In contrast, the community hospital's COVID-19 step-down unit only wore N95/KN95 masks—these yielded only negative qPCR results.

Rates of positivity varied by role of the PPE wearer. Bedside RNs, patients wearing the PPE, and the surgical technician, all had samples that tested positive. The anesthesiologist, dialysis RNs, RN circulator in the operating room, and respiratory therapist, did not have any exposed PPE that tested positive by qPCR (Table 2).

We compared the time between the patient's last recorded positive COVID-19 test and date of the collection of a PPE that yielded positive qPCR tests to determine if any relationships existed, however, we did not find significant consistencies (Fig 2).

### Laboratory setting

The control experiment yielded higher viral counts for the majority of droplet simulated samples during both time intervals than for the nebulized samples. Total viral counts were

**Table 3. Positives over time by exposure.**

| Exposure Type | Total Samples | N positive | % positive (95% CI) |
|---|---|---|---|
| *Direct Cough* | | | |
| *30 Minutes* | 12 | 1 | 8.3% (0.0% - 37.5%) |
| *12 Hours* | 12 | 1 | 8.3% (0.0% - 37.5%) |
| *HFNC* | | | |
| *30 Minutes* | 66 | 4 | 6.1%(1.9% - 15.0%) |
| *12 Hours* | 63 | 4 | 6.3%(2.0% - 15.7%) |
| **Total** | | | |
| *30 Minutes* | 78 | 5 | 6.4%(2.4% - 14.5%) |
| *12 Hours* | 75 | 5 | 6.7%(2.5% - 15.0%) |

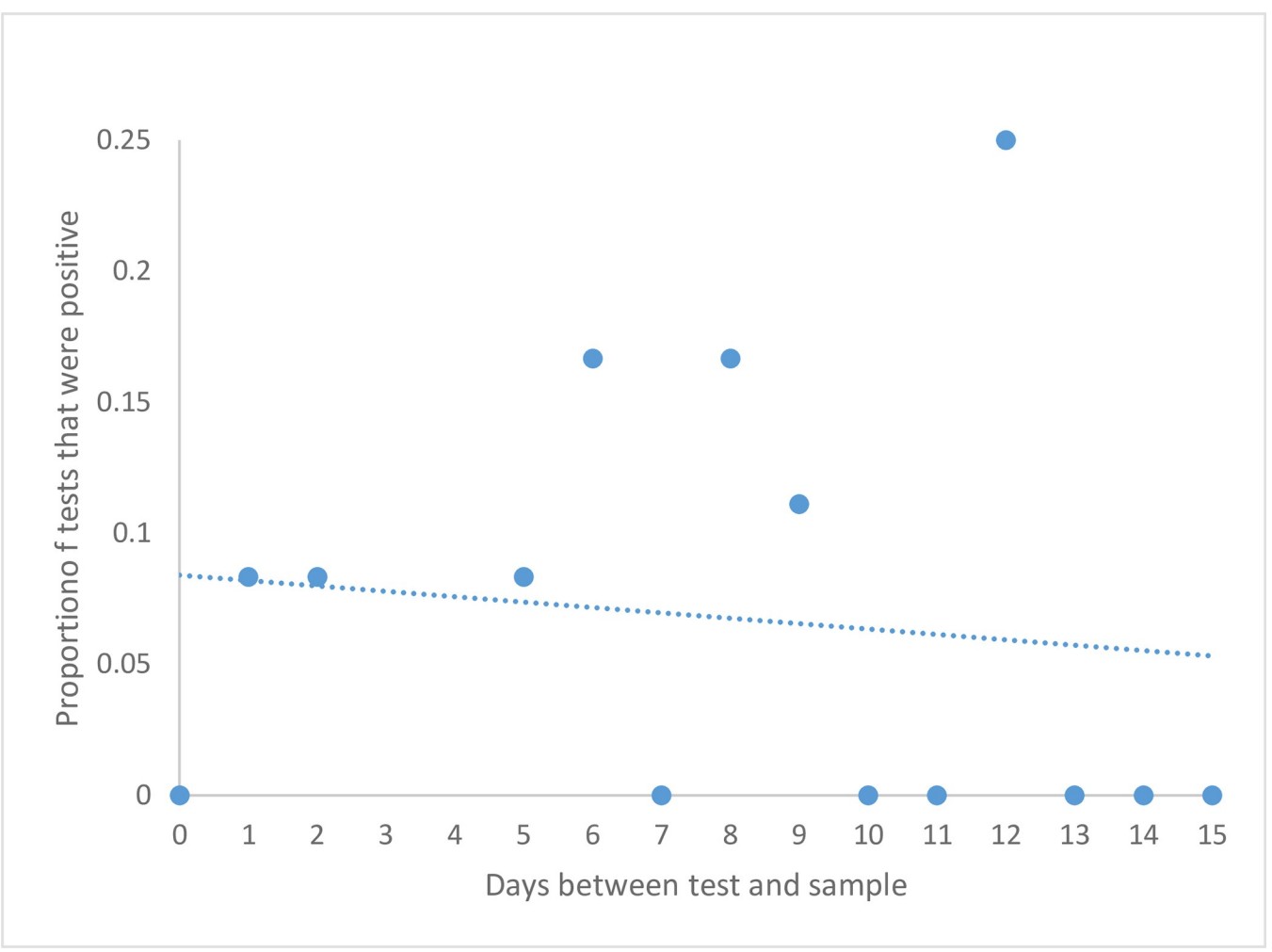

**Fig 2. Positive samples association from date of PCR test.** The time between the collection date of the PPE sample and the patient's last positive COVID-19 PCR test was compared to the percent of positive PPE and bag samples (total).

significantly different between droplet and nebulized samples (Wilcoxon signed rank test, p = 0.001). Median count for droplet was 924 and 112 for nebulized (Fig 3). There was not enough evidence to show a significant difference in count between the 30 minute and 12-hour measurements (p = 0.140). There appear to be marked mean abundance differences by duration (median for 30 minutes is 634 versus 12-hour median is 288), and a statistically significant may have emerged with increased sample size.

## Discussion

As the pandemic continues, and future PPE supply limitations remain precarious, the investigation on storage receptacles for the reuse of PPE is critical for protecting frontline HCWs. This is one of the first studies to examine the potential for fomite transfer of SARS-CoV-2 viral particles from PPE to storage receptacles, paper bags, for the reuse of HCW PPE. Even after high-risk exposure to AGPs, most of the samples we collected (93.2%) did not test positive for SARS-CoV-2 after either time interval.

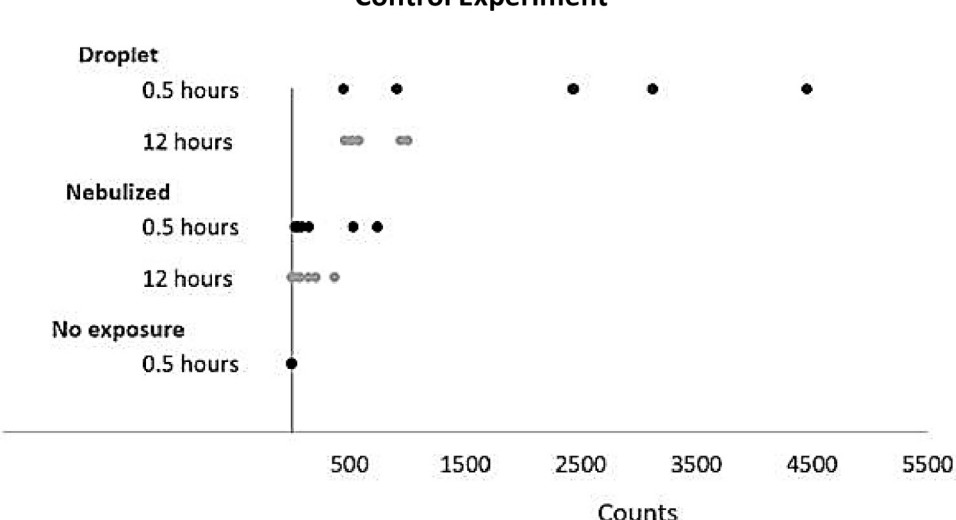

**Fig 3. Control experiment.** Inactivated SARS-CoV-2 was nebulized to simulate aerosol events or placed by pipette to simulate droplet events to determine absorption and recovery from brown paper bag swatches and compared with samples without exposure. (Counts = total viral counts recovered on PCR).

Although both community and tertiary care centers had patients with SARS-CoV-2, positive swabs were found only at the tertiary care center (paper bags and PPE). Specifically, they were found more often in locations in the COVID-19 step-down unit, 16.7% compared to 8.8% in the ICU, and 4.8% in the emergency department. Potential causes for site differences could be due to protocols on how to correctly re-use PPE, deviation from those protocols, or poor staff-to-patient ratios (in step down unit compared to ICU) whereby there is overall protocol fatigue and non-intentional relaxing of safety measures. While the abundance of virus found on tested PPE samples was low, some were positive, and such significant site differences needs further investigation as protocols could be improved to replicate optimal environments where there is low or no positivity samples.

Our results demonstrate that recoverable SARS-CoV-2 RNA on exposed PPE items collected varied by procedure. The highest positivity rates were associated with ventilator disconnection (n = 6, 16.7% positivity), followed by exposure to respiratory droplets from coughing (n = 24, 8.3% positivity) and exposure to high-flow nasal cannula (n = 129, 6.2% positivity). Tracheostomy surgery had the lowest positivity rate (n = 18, 5.6% positivity). This is consistent with recent findings that show ventilator disconnection causes high rates of contamination and high flow nasal cannula to be contaminated at a lower rate [8,20,21]. In a situation where PPE supply is slightly improved, guidelines could be created to reuse only for certain types of exposure, such as no re-use for exposure to ventilator disconnection.

Interestingly, contamination of SARS-CoV-2 was found on the *unexposed* external surfaces of the paper bag and on the *unexposed* surface of the PPE. We hypothesize that asymptomatic COVID-19 positive HCW may have contaminated this surface and be a source of fomite transmission or other environmentally mediated transfer. Although quantities were found to be low, the infectious dose for SARS-CoV-2 remain uncertain [1,8]. This is a potential source of fomite transfer that has not been well studied and requires further investigation.

Additionally, our positive controls using a nebulizer to simulate real-world aerosolized SARS-CoV-2 demonstrated limited detection after 30 minutes and even lower detection after 12-hours. However, the droplet samples containing SARS-CoV-2 yielded higher viral counts

than the nebulized samples following 30 minutes, demonstrating the *potential* for transfer from the paper bag to the individual. Encouragingly, amongst the controls, sampling at the 12-hour time point demonstrated a decreasing trend in the detectable level of SARS-CoV-2 RNA. Although the positive control demonstrated that SARS-CoV-2 RNA can be recovered from the paper bag, the reduced RNA load at these time intervals decreases the likelihood that enough viral transfer, if any, will occur that could lead to infection. These results suggest a low probability of fomite transfer, and confirm assertions from previous investigations [22,23]. Our results show that secondary contact with contaminated PPE and repeat use of stored PPE is unlikely to be a significant source of exposure [23–25].

## Limitations

This study was primarily limited by the restricted supply of PPE available to HCWs and researchers early in the pandemic. At the time, HCWs were required to wear their N95/KN95 masks for their entire shift and only had access to one face shield each making it difficult for research staff to test PPE at 12-hour intervals. We accounted for this by limiting our collection to PPE worn in a patient room with an AGP >20 minutes within an hour of collection at the end of the shift. This study used surface swabs for sampling. It is possible that surface swabbing may be insufficient for the detection of entrapped viral particles from these substrates.[16] HCWs COVID-19 status was unknown during the study, and testing was limited at the time.

## Conclusion

During the beginning of the COVID-19 pandemic, the CDC recommended guidelines for storage and re-use of personal protective equipment for hospitals experiencing critical shortages [3]. These guidelines were unstudied and based on indirect evidence [6,10]. With the persistence of the Delta variant and rise of the Omicron-variant and future SARS-COV-2 variants, PPE shortages will be a continuous impending threat to healthcare workers at the front lines. Thus, our investigation confirmed that paper bags containing PPE exposed to SARS-CoV-2 can have contamination with detectable viral RNA after 30 minutes and 12 hours after exposure. This contamination was primarily seen on the outside surfaces of the paper bag and on the unexposed surface of the PPE item, a potential source of fomite transmission that needs further investigation. Our results however indicate that the likelihood of paper bags being a significant source of disease transmission is unlikely. We conclude that storing PPE in a paper bag for re-use is a recognized safe practice and that secondary contact from contaminated PPE or repeated use is unlikely to be a significant source of exposure. Similar studies should be repeated for each infectious agent during times of limited resources when new protocols for reuse are needed. It is critical that improved guidelines on PPE re-use are needed and replicating practice environments with low contamination rates may prevent future transmission events.

## Author Contributions

**Conceptualization:** Kyirsty Unger, Bory Kea.

**Data curation:** Kyirsty Unger, Leslie Dietz, Patrick Horve, Amber Lin, Bory Kea.

**Formal analysis:** Kyirsty Unger, Leslie Dietz, Patrick Horve, Kevin Van Den Wymelenberg, Amber Lin, Bory Kea.

**Funding acquisition:** Kyirsty Unger, Leslie Dietz, Patrick Horve, Kevin Van Den Wymelenberg, Bory Kea.

**Investigation:** Kyirsty Unger, Leslie Dietz, Patrick Horve, Kevin Van Den Wymelenberg, Bory Kea.

**Methodology:** Kyirsty Unger, Leslie Dietz, Patrick Horve, Kevin Van Den Wymelenberg, Bory Kea.

**Project administration:** Kyirsty Unger, Bory Kea.

**Resources:** Kevin Van Den Wymelenberg, Bory Kea.

**Supervision:** Kyirsty Unger, Bory Kea.

**Validation:** Kyirsty Unger, Leslie Dietz, Patrick Horve, Amber Lin, Bory Kea.

**Visualization:** Kyirsty Unger, Bory Kea.

**Writing – original draft:** Kyirsty Unger, Leslie Dietz, Patrick Horve, Kevin Van Den Wymelenberg, Amber Lin, Bory Kea.

**Writing – review & editing:** Kyirsty Unger, Leslie Dietz, Patrick Horve, Kevin Van Den Wymelenberg, Amber Lin, Erin Kinney, Bory Kea.

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
