## [Decision Letter · Decision Letter 0]

30 Jun 2022

PONE-D-22-12653Evaluating Fomite Risk of Brown Paper Bags Storing Personal Protective Equipment Exposed to SARS-CoV-2: A Quasi-Experimental StudyPLOS ONE

Dear Dr. Unger,

Thank you for submitting your manuscript to PLOS ONE. After careful consideration, we feel that it has merit but does not fully meet PLOS ONE’s publication criteria as it currently stands. Therefore, we invite you to submit a revised version of the manuscript that addresses the points raised during the review process.

Please address all the comment from Reviewers 1, 2 and 3 to approach some weaknesses of your work. 

We look forward to receiving your revised manuscript.

Kind regards,

Celia Andreu-Sánchez

Academic Editor

PLOS ONE

Journal Requirements:

2. You indicated that ethical approval was not necessary for your study. We understand that the framework for ethical oversight requirements for studies of this type may differ depending on the setting and we would appreciate some further clarification regarding your research. Could you please provide further details on why your study is exempt from the need for approval and confirmation from your institutional review board or research ethics committee (e.g., in the form of a letter or email correspondence) that ethics review was not necessary for this study? Please include a copy of the correspondence as an "Other" file.

"Yes- this project was funded by an internal institution COVID-19 rapid response grant from Oregon Health and Science University."

"BK - is funded by an NHBLI RO1HL 57598.

KV - has a company called Duktile through which he provides healthy building consulting, including consulting related to viral pathogens, and he serves as a scientific advisor to EnviralTech, a company that conducts viral environmental surveillance, including in senior care facilities."

Reviewers' comments:

Reviewer's Responses to Questions

**Comments to the Author**

1. Is the manuscript technically sound, and do the data support the conclusions?

Reviewer #1: Yes

Reviewer #2: Yes

Reviewer #3: Yes

2. Has the statistical analysis been performed appropriately and rigorously? 

Reviewer #1: I Don't Know

Reviewer #2: Yes

Reviewer #3: N/A

3. Have the authors made all data underlying the findings in their manuscript fully available?

Reviewer #1: Yes

Reviewer #2: No

Reviewer #3: Yes

4. Is the manuscript presented in an intelligible fashion and written in standard English?

Reviewer #1: Yes

Reviewer #2: Yes

Reviewer #3: Yes

5. Review Comments to the Author

Reviewer #1: Overall Comments:

Thank you for the opportunity to review your work! Overall, I think this is a well written paper and an important investigation that has a clear benefit to informing protocols for used PPE storage. I have some major and minor comments to consider to add clarity and context in some places for readers.

Major Comments:

What was the swab efficiency? This will have important implications for interpreting % positive.

The objective of the Controls portion of the study is a little unclear. Is this related to contamination of bags due to airborne virus and evaluating whether this could be related to fomite transmission from contaminated bags as opposed to virus from contaminated PPE?

In the Controlled experiment, can you explain how the steady state concentration was estimated? What is the air exchange rate in the room, and how was this determined? Was the air filtered after experiments? What was the temperature and relative humidity, and this will affect aerosol sizes and the distances they travel?

In the statistical analysis, could Fisher’s exact tests be used to compare % positive across PPE type, exposure type, and location separately to look for statistically significant differences? That might be a nice addition to Table 1.

Related to a above comment, were there negative controls or field controls in addition to sampling the area for environmental contamination (in the environmental study, not the controlled study)? In other words, could the bags be contaminated with SARS-CoV-2 due to exposure to the air (aerosols or droplets) as opposed to or in addition to the PPE in the real-world environment? In the Discussion where the detection of SARS-CoV-2 on unexposed external surfaces of the paper bag is discussed, I think this is worth considering/mentioning. If this was not done, perhaps it is a limitation that could be mentioned – some insights into potential contamination via airborne or droplet spray in the controlled study but not in a real-world environment.

What was the limit of detection?

Figure 2 is interesting. If you stratify by PPE type or the type of aerosolizing procedure (or other potential confounders like facility type), do you see anything different? There appears to be a strong linear relationship for a certain set and no relationship with another.

In the Discussion where the controls are discussed, I think it’s important to distinguish between swab efficiency and transfer efficiency. The transfer efficiency of a gloved or ungloved touch of a bag is likely to be different than swabbing. Maybe a quick back-of-the-envelope example would be useful. Something like, “If we assume a transfer efficiency of ______% and ______ gc/cm^2 on the bag, only ______ gc would transfer to the fingertip (assuming 2 cm^2 of fingertip).”

Minor Comments:

In the Introduction where it’s said that HCWs face fomite transmission risk from “prolonged use” of PPE, I’m not sure what this means. Are you referring to things like hand-to-mask contact and, later, hand-to-face contact?

I like the Introduction – nice lead up to the objectives with a clear stated impact of the work, as well. The importance of fomite transmission is also not overstated, which I think is appropriate.

I think there’s a typo in the first subheading under Methods

Can the geographical location of the two regional urban hospitals be confirmed in the text? This might help if others do systematic literature reviews that include yours in their search.

I’m not sure what this means: “Surgical masks (ATSM level 3 mask) worn for over 20 minutes by a symptomatic hospitalized patient with a positive COVID-19 PCR test were included.” – Included as an AGP or included as samples? How are these samples related to the focus on PPE for healthcare?

Where it says all PPE was doffed and cleaned, does this refer to disinfection or just cleaning, such as wiping of with a non-disinfectant chemical?

Isn’t SARS-CoV-3 designated as BSL-3?

Figure 3 – Is it absorption of virus in the swab eluent or adsorption of the virus to the bag surface?

Reviewer #2: General Comments:

The work evaluates how effective of a barrier for fomite transmission of SARS-CoV-2 are paper bags storing masks, respirators, and face shields. Sampling (every 0.5 and 12 hours) for the presence of SARS-CoV-2 took place in the interior and exterior surfaces of paper bags. The processing included 177 swabs from 30 PPE samples. A low ~7% positivity rate was found among all samples and associated to ventilator disconnection causing exposure to bioaerosol from coughing. In conclusion, paper bags were not a significant fomite risk for SARS-CoV-2 transmission, suggesting that storing and re-using PPE when demanded with urgency is possible. Although the manuscript is generally well written and the methods and analysis appears appropriate, there are three major issues that should be addressed during a revision before the manuscript could be accepted for publication.

Major Comments:

1) The revised manuscript should increase the front size of Figure 1 to become readable after final professional editing for publication. It looks too small in the current form. Also, the current format of Figures 2 and 3 is not acceptable for publication and the figures should be recreated by plotting the data and exporting them with high resolution. The gray surrounding background should also be removed from Figure 2. Is there any statistical treatment for Figure 2?

2) For the Introduction section, the research is missing a connection to a key paper from the Health Planning and Management field that has escaped the attention of the authors but is facilitated here. The manuscript should have explained the work of Guzman (An overview of the effect of bioaerosol size in coronavirus disease 2019 transmission. Int J Health Plann Mgmt 2021, 36: 257-266. DOI: 10.1002/hpm.3095).

3) The conclusions are simply repeating statements summarizing the data that can remain in the revision. However, the manuscript should incorporate in the revision a deeper analysis of the outcomes of this work at the global level, for different settings and countries.

Reviewer #3: I would like to thank the journal and authors for the opportunity of revising the manuscript "Evaluating Fomite Risk of Brown Paper Bags Storing Personal Protective Equipment Exposed to SARS-CoV-2: A Quasi-Experimental Study".

The manuscript can be accepted for publication.

6. PLOS authors have the option to publish the peer review history of their article (what does this mean?). If published, this will include your full peer review and any attached files.

Reviewer #1: No

Reviewer #2: No

Reviewer #3: No

---

## [Author Response · Author response to Decision Letter 0]

29 Jul 2022

Here is a point-by-point response to the reviewers’ comments and concerns.

Reviewer #1: Overall Comments:

Thank you for the opportunity to review your work! Overall, I think this is a well written paper and an important investigation that has a clear benefit to informing protocols for used PPE storage. I have some major and minor comments to consider to add clarity and context in some places for readers.

Response: Thank you very much for agreeing with us on the intention of this manuscript. We have read your comments carefully and tried our best to address them one by one. We hope that the manuscript has been improved towards PLOS ONE standards after this revision.

Major Comments:

Comment 1:What was the swab efficiency? This will have important implications for interpreting % positive.

The objective of the Controls portion of the study is a little unclear. Is this related to contamination of bags due to airborne virus and evaluating whether this could be related to fomite transmission from contaminated bags as opposed to virus from contaminated PPE?

Response: Thank you for this comment. You have raised an important point here. We have relied on the swab efficiency evaluated by the R&D of the company that manufactures them. For example, A study by WHO early in the pandemic evaluated the performance of six different swabs and found that a synthetic material tipped swab was superior for retaining the most virus and for not interfering with downstream lab analysis. Here is that paper: https://www.ncbi.nlm.nih.gov/pmc/articles/PMC7414358/ We do not believe there is a need for us to retest the efficiency of these swabs. However, we agree that it is important to give clarification to this and we have referenced the above study in our manuscript. 

Comment 2: In the Controlled experiment, can you explain how the steady state concentration was estimated? What is the air exchange rate in the room, and how was this determined? Was the air filtered after experiments? What was the temperature and relative humidity, and this will affect aerosol sizes and the distances they travel?

Response: Thank you for requesting this clarification. The following statements were added to the control section of our paper:

In the control experiment, a known concentration of heat-inactivated SARS-CoV-2 was nebulized. This was prepared from a concentrated stock that had been quantified by the BSL3 lab at Montana State University where it was cultivated and attenuated and then verified by qPCR by our lab upon arrival. 

 The control experiment, nebulization were conducted in a sealed room-scale controlled environment with an internal volume of 28,040 L with an internal temperature maintained at 22°C +/- 4°C and a relative humidity maintained at 50% +/- 10% with the use of a single portable humidifier. (Horve et al 2021) Air was circulated in the room using two oscillating fans, which moved 24,975 L of air per minute. (Horve et al 2021) During this experiment, the air exchange rate was 1 ACH and was regulated and sustained using a timed operation HEPA exhaust, with make up air via infiltration (Horve et al 2021).

Source: https://journals.plos.org/plosone/article?id=10.1371/journal.pone.0257689

Comment 3: In the statistical analysis, could Fisher’s exact tests be used to compare % positive across PPE type, exposure type, and location separately to look for statistically significant differences? That might be a nice addition to Table 1.

Response: Thank you for this question, we considered many relationships with this data. We concluded that these samples are inherently clustered, so we would need to account for that. However, there was not enough sample size and enough positivity to make any sort of modeling work. Given the % CIs nothing would be significant. Table 1 is more of a descriptive table displaying the number of samples and where they were from so we would be unable to add p-values. 

Comment 4: Related to a above comment, were there negative controls or field controls in addition to sampling the area for environmental contamination (in the environmental study, not the controlled study)? In other words, could the bags be contaminated with SARS-CoV-2 due to exposure to the air (aerosols or droplets) as opposed to or in addition to the PPE in the real-world environment? In the Discussion where the detection of SARS-CoV-2 on unexposed external surfaces of the paper bag is discussed, I think this is worth considering/mentioning. If this was not done, perhaps it is a limitation that could be mentioned – some insights into potential contamination via airborne or droplet spray in the controlled study but not in a real-world environment.

Response: Thank you for your comment, we agree with you that this is worth mentioning and have added clarification to our manuscript in the Sample Collection heading. The paper bags that were obtained for testing were kept in a sealed bag until the time of testing. At the time of experimentation, the brown paper bags were taken out of the sterile sealed bag and then placed in the testing room. After nebulizing, swab samples were collected and placed in designated tubes for processing. Additionally, extraction controls were conducted with each batch to confirm the integrity of the cleanliness of the brown paper bags. 

Comment 5: What was the limit of detection?

Response: Thank you for your question. We assume this is regarding the TaqPath assay itself – if so, it is LoD is 10 gc equivalents. 

“What is the sensitivity and specificity of the TaqPath COVID-19 Combo kits? You can find a detailed writeup of the performance characteristics used to assess analytical performance and clinical performance of the TaqPath COVID-19 Combo Kit in the Instructions for Use. These include: limit of detection (10 genomic copy equivalents (GCE)/reaction), reactivity (homology of assay designs to known SARS-CoV-2 genomes), interfering substances analysis (no false positive results were observed for any of the substances and concentrations tested), cross-reactivity (in silico analysis of 43 organisms for potential cross-reactivity or interference), and clinical evaluation of 60 contrived positive and 60 negative specimens to evaluate kit performance. These characteristics taken together help to confirm that the test performs as expected.”

Source: https://assets.thermofisher.com/TFS-Assets/GSD/Reference-Materials/taqpath-covid-19-eua-faq.pdf

Comment 6: Figure 2 is interesting. If you stratify by PPE type or the type of aerosolizing procedure (or other potential confounders like facility type), do you see anything different? There appears to be a strong linear relationship for a certain set and no relationship with another.

Response: Thank you for this question. We considered subgroup analysis. Due to the sample size, subgroup analyses were not feasible.

Comment 7: In the Discussion where the controls are discussed, I think it’s important to distinguish between swab efficiency and transfer efficiency. The transfer efficiency of a gloved or ungloved touch of a bag is likely to be different than swabbing. Maybe a quick back-of-the-envelope example would be useful. Something like, “If we assume a transfer efficiency of ______% and ______ gc/cm^2 on the bag, only ______ gc would transfer to the fingertip (assuming 2 cm^2 of fingertip).”

Response: Thank you for this comment and I think you bring up interesting questions. However, It was not in the scope of this study to explore the transfer efficiency of the swabs used. The viral transfer of different swabs has been documented and our lab has considered these studies and used them to guide our swab and media choices. 

Sources: https://journals.asm.org/doi/full/10.1128/JCM.01562-20, https://www.ncbi.nlm.nih.gov/pmc/articles/PMC7414358/, https://www.mdpi.com/2075-4418/12/1/206/htm

Minor Comments:

Comment 8: In the Introduction where it’s said that HCWs face fomite transmission risk from “prolonged use” of PPE, I’m not sure what this means. Are you referring to things like hand-to-mask contact and, later, hand-to-face contact?

Response: Thank you for this comment. We agree that the statement above requires clarification. When our study began, we were investigating the CDC recommendations for prolonged use for PPE and limited reuse. These practices were no longer a standard by the end of our data collection. We simplified the original language to apply it to a broader context. We have added the following clarification to our manuscript:

Prolonged use is defined as any application of PPE worn continuously beyond standard patient care or greater than 1 hour of continuous use. 

Source: https://blogs.cdc.gov/niosh-science-blog/2020/06/10/ppe-burden/

Comment 9: I like the Introduction – nice lead up to the objectives with a clear stated impact of the work, as well. The importance of fomite transmission is also not overstated, which I think is appropriate.

Response: Thank you for the feedback.

Comment 10: I think there’s a typo in the first subheading under Methods

Response: This has been deleted. Thank you. 

Comment 11: Can the geographical location of the two regional urban hospitals be confirmed in the text? This might help if others do systematic literature reviews that include yours in their search.

Response: We have added clarification in our manuscript:

“This is a quasi-experimental study. Samples were collected from two regional urban hospitals in the Portland metropolitan area.”

Comment 12: I’m not sure what this means: “Surgical masks (ATSM level 3 mask) worn for over 20 minutes by a symptomatic hospitalized patient with a positive COVID-19 PCR test were included.” – Included as an AGP or included as samples? How are these samples related to the focus on PPE for healthcare?

Response: Thank you for this question. This sample was not included as an AGP, but was included under our sampling as a potential droplet exposure. We believe it relates to health care workers because surgical masks can be exposed to COVID and be re-used multiple times. 

Comment 13: Where it says all PPE was doffed and cleaned, does this refer to disinfection or just cleaning, such as wiping of with a non-disinfectant chemical?

Response: Thank you for the question. We have clarified our procedure by PPE type in our manuscript. 

“Gowns were doffed and discarded. Face shields and eye protection were cleaned with Oxivir. The PPE item to be sampled was placed in a clean brown paper bag and placed in a designated sampling area.”

Comment 14: Isn’t SARS-CoV-3 designated as BSL-3?

Response: Thank you for this question. SARS-CoV-2 is a BSL-3 organism. The SARS-CoV-2 we nebulize for controls has been heat-inactivated and is not infective. Furthermore, the samples we take are placed in DNA/RNA Shield, a lysis buffer which inactivates all organisms placed in the buffer.

Comment 15: Figure 3 – Is it absorption of virus in the swab eluent or adsorption of the virus to the bag surface?

Response: Figure 3 shows all the swabs from the control experiment, with the y axis being time and the x axis being gene copies per ul. There is no absorption being measured. We have adjusted the description of the graph for clarification. 

Reviewer #2: 

General Comments:

The work evaluates how effective of a barrier for fomite transmission of SARS-CoV-2 are paper bags storing masks, respirators, and face shields. Sampling (every 0.5 and 12 hours) for the presence of SARS-CoV-2 took place in the interior and exterior surfaces of paper bags. The processing included 177 swabs from 30 PPE samples. A low ~7% positivity rate was found among all samples and associated to ventilator disconnection causing exposure to bioaerosol from coughing. In conclusion, paper bags were not a significant fomite risk for SARS-CoV-2 transmission, suggesting that storing and re-using PPE when demanded with urgency is possible. Although the manuscript is generally well written and the methods and analysis appears appropriate, there are three major issues that should be addressed during a revision before the manuscript could be accepted for publication.

Response: Thank you for taking the time to read and comment on our manuscript. We have read your comments carefully and tried our best to address them.

Major Comments:

Comment 1: The revised manuscript should increase the front size of Figure 1 to become readable after final professional editing for publication. It looks too small in the current form. Also, the current format of Figures 2 and 3 is not acceptable for publication and the figures should be recreated by plotting the data and exporting them with high resolution. The gray surrounding background should also be removed from Figure 2. Is there any statistical treatment for Figure 2?

Response: Thank you for this comment. As you suggested, we have adjusted the font for Figure 1 and reformatted figures 1 and 3 according to the PLOS ONE standards. We considered adding a linear trend through figure 2, but the slope would be close to zero so the addition would not be beneficial. 

Comment 2: For the Introduction section, the research is missing a connection to a key paper from the Health Planning and Management field that has escaped the attention of the authors but is facilitated here. The manuscript should have explained the work of Guzman (An overview of the effect of bioaerosol size in coronavirus disease 2019 transmission. Int J Health Plann Mgmt 2021, 36: 257-266. DOI: 10.1002/hpm.3095).

Response: Thank you for bringing this to our attention. We agree that the article presented does indeed add to the literature surrounding COVID-19 transmission rates and will be an informative source for our paper. At the time of our study’s design, this had not been published so it was not used for the creation of our experiment protocols. We have added it as a source for our introduction. Guzman’s findings do provide important clarification on the primary transmission routes for SARS-CoV-2. 

Comment 3: The conclusions are simply repeating statements summarizing the data that can remain in the revision. However, the manuscript should incorporate in the revision a deeper analysis of the outcomes of this work at the global level, for different settings and countries.

Response: Thank you for your comment. While we emphasize our study is encouraging for safety for HCW. We also will emphasize that hypothesizing and setting new guidelines for the reuse of PPE without data is insufficient. Similar studies should be repeated for each infectious agent during times of limited resources when new protocols for reuse are needed.

Reviewer #3: I would like to thank the journal and authors for the opportunity of revising the manuscript "Evaluating Fomite Risk of Brown Paper Bags Storing Personal Protective Equipment Exposed to SARS-CoV-2: A Quasi-Experimental Study".

The manuscript can be accepted for publication.

Response: Thank you for your time and feedback.

---

## [Decision Letter · Decision Letter 1]

9 Aug 2022

Evaluating Fomite Risk of Brown Paper Bags Storing Personal Protective Equipment Exposed to SARS-CoV-2: A Quasi-Experimental Study

PONE-D-22-12653R1

Dear Dr. Unger,

We’re pleased to inform you that your manuscript has been judged scientifically suitable for publication and will be formally accepted for publication once it meets all outstanding technical requirements.

Kind regards,

Celia Andreu-Sánchez

Academic Editor

PLOS ONE

Additional Editor Comments (optional):

Reviewers' comments:

Reviewer's Responses to Questions

**Comments to the Author**

1. If the authors have adequately addressed your comments raised in a previous round of review and you feel that this manuscript is now acceptable for publication, you may indicate that here to bypass the “Comments to the Author” section, enter your conflict of interest statement in the “Confidential to Editor” section, and submit your "Accept" recommendation.

Reviewer #1: All comments have been addressed

Reviewer #2: All comments have been addressed

2. Is the manuscript technically sound, and do the data support the conclusions?

Reviewer #1: Yes

Reviewer #2: Yes

3. Has the statistical analysis been performed appropriately and rigorously? 

Reviewer #1: Yes

Reviewer #2: Yes

4. Have the authors made all data underlying the findings in their manuscript fully available?

Reviewer #1: Yes

Reviewer #2: Yes

5. Is the manuscript presented in an intelligible fashion and written in standard English?

Reviewer #1: Yes

Reviewer #2: Yes

6. Review Comments to the Author

Reviewer #1: Thank you for adequately responding to my comments and providing thoughtful responses. I don't have any further comments.

Reviewer #2: The work evaluates how effective of a barrier for fomite transmission of SARS-CoV-2 are paper bags storing masks, respirators, and face shields. Sampling (every 0.5 and 12 hours) for the presence of SARS-CoV-2 took place in the interior and exterior surfaces of paper bags. The processing included 177 swabs from 30 PPE samples. A low ~7% positivity rate was found among all samples and associated to ventilator disconnection causing exposure to bioaerosol from coughing. In conclusion, paper bags were not a significant fomite risk for SARS-CoV-2 transmission, suggesting that storing and re-using PPE when demanded with urgency is possible. The work has been improved after all comments from a previous version have been addressed. The work is recommended for publication in PLOS ONE.

7. PLOS authors have the option to publish the peer review history of their article (what does this mean?). If published, this will include your full peer review and any attached files.

Reviewer #1: No

Reviewer #2: No

---

## [Editor Report · Acceptance letter]

16 Aug 2022

PONE-D-22-12653R1 

Evaluating Fomite Risk of Brown Paper Bags Storing Personal Protective Equipment Exposed to SARS-CoV-2: A Quasi-Experimental Study 

Dear Dr. Unger:

I'm pleased to inform you that your manuscript has been deemed suitable for publication in PLOS ONE. Congratulations! Your manuscript is now with our production department. 

Kind regards, 

on behalf of

Dr. Celia Andreu-Sánchez 

Academic Editor

PLOS ONE